# Technical Advances in Segmentectomy for Lung Cancer: A Minimally Invasive Strategy for Deep, Small, and Impalpable Tumors

**DOI:** 10.3390/cancers13133137

**Published:** 2021-06-23

**Authors:** Takashi Eguchi, Toshihiko Sato, Kimihiro Shimizu

**Affiliations:** 1Division of General Thoracic Surgery, Department of Surgery, School of Medicine, Shinshu University, Matsumoto 390-8621, Japan; eguchi_t@shinshu-u.ac.jp; 2Department of General Thoracic, Breast, Pediatric Surgery, Faculty of Medicine, Fukuoka University, Fukuoka 814-0180, Japan; oshisato@fukuoka-u.ac.jp

**Keywords:** localization, lung segmentectomy, radiofrequency identification, three-dimensional computed tomography

## Abstract

**Simple Summary:**

The use of minimally invasive lung segmentectomy for early-stage lung cancer is increasing. This procedure is associated with technical challenges because (1) it requires a thorough understanding of the complex segmental anatomy that frequently accompanies anomalies, and (2) it is difficult to confirm the location of small tumors during minimally invasive surgery, which makes it difficult to obtain adequate surgical margins. Herein, we summarize the published evidence and discuss key issues related to minimally invasive segmentectomy. We focus on overall efforts to overcome these challenges, including preoperative planning and simulation for segmentectomy, and intraoperative localization of small tumors.

**Abstract:**

With the increased detection of early-stage lung cancer and the technical advancement of minimally invasive surgery (MIS) in the field of thoracic surgery, lung segmentectomy using MIS, including video- and robot-assisted thoracic surgery, has been widely adopted. However, lung segmentectomy can be technically challenging for thoracic surgeons due to (1) complex segmental and subsegmental anatomy with frequent anomalies, and (2) difficulty in localizing deep, small, and impalpable tumors, leading to difficulty in obtaining adequate margins. In this review, we summarize the published evidence and discuss key issues related to MIS segmentectomy, focusing on preoperative planning/simulation and intraoperative tumor localization. We also demonstrate two of our techniques: (1) three-dimensional computed tomography (3DCT)-based resection planning using a novel 3DCT processing software, and (2) tumor localization using a novel radiofrequency identification technology.

## 1. Introduction

With rapid development and innovation in the field of oncology, including diagnostic modalities, medical/radiological treatments, and genomic knowledgebases, the basis of cancer treatment has been shifting from a “one-drug (treatment)-fits-all approach” to a “precision medicine” approach, which is a treatment tailored to individual patients. However, for thoracic surgeons, lobectomy has been the evidence-based standard surgical treatment for early-stage lung cancer since 1995 when a randomized control trial (LCSG 821) compared lobectomy and sublobar resection (including segmentectomy and wedge resection), demonstrating the former’s prognostic superiority [1].

Compared with other solid tumors, lung cancer carries a relatively high risk of both cancer-related and non-cancer-related mortality because of the increase in comorbidities with older age [2,3]. With the recent prevalence of lung cancer screening using computed tomography (CT), the detection of tumors <2 cm is increasing, and consequently, the prognosis of small lung cancer has improved [4]. Despite concerns regarding the oncological adequacy of sublobar resection, its use has been increasing, especially in older patients with comorbidities [5]. Moreover, sublobar resection, particularly segmentectomy, is considered a definitive surgery for small lung cancers, with non-inferior prognosis and better postoperative lung function than lobectomy [6]. Recently, the results of a large, multi-institutional, prospective randomized trial assessing the outcomes of segmentectomy and lobectomy (JCOG0802) were demonstrated (the 101st Annual Meeting of the American Association of Thoracic Surgery, 30 April–2 May, 2021); the overall survival of patients who underwent segmentectomy was significantly superior to that of patients who underwent lobectomy, suggesting that segmentectomy is a potentially definitive, standard surgical approach in patients with small, early stage lung cancers.

The use of thoracic minimally invasive surgery (MIS), including video-assisted thoracic surgery (VATS) and robot-assisted thoracic surgery (RATS), has increased with the growing evidence of improved outcomes, such as postoperative pain, quality of life, complications, and non-cancer-related mortality [7,8,9,10]. A recent study using the United States national database demonstrated that the use of MIS segmentectomy has been increasing without compromising oncologic outcomes, compared with the open approach for clinical stage IA lung cancer [11].

Lung segmentectomy is a technically challenging procedure for thoracic surgeons because of the complex segmental and subsegmental anatomy with frequent anomalies [12,13], as well as the difficulty in localizing tumors, especially in cases with deep, small, and impalpable tumors, which make it difficult to obtain adequate margins. To overcome these challenges, several studies have investigated lung segmental anatomy and preoperative simulation on the basis of individual anatomy by three-dimensional (3D) CT [14,15,16,17,18,19], efforts to localize tumors intraoperatively [20,21,22], and identify the intersegmental plane [23,24].

Herein, we summarize the published evidence and discuss key issues related to MIS segmentectomy, focusing on preoperative simulation, resection planning, and intraoperative tumor localization. We also demonstrate two of our techniques: (1) 3DCT-based resection planning using a novel 3DCT processing software, and (2) tumor localization using a novel radiofrequency identification (RFID) technology.

## 2. Minimally Invasive Segmentectomy

Despite wide variations in lung segmentectomy, they can be classified into simple (typical) and complex (atypical) [25,26]. Simple segmentectomy includes segmentectomies for S6 (superior segment) or basilar segments in the lower lobes, and upper division (tri-segment) or lingula in the left upper lobe, in which only one linear intersegmental plane is divided. Complex segmentectomy includes all other segmentectomies in which more than one intersegmental plane is divided (individual segments in the upper lobe or basilar segment [S1, S2, S1 + 2, S3, S7, S8, S9, S10], or includes segments that are rarely indicated for segmentectomy, such as segments in the middle lobe [27,28,29]. In addition to the difficulty in treating more than one intersegmental plane, hilar dissection is generally more challenging in complex segmentectomies than in simple segmentectomies. In general, the MIS approach for segmentectomies, particularly complex segmentectomies, requires greater skill than the open approach. Therefore, simple MIS segmentectomies have been relatively widely performed compared to complex MIS segmentectomies [25,30].

### 2.1. Minimally Invasive Techniques for Complex Segmentectomy

In 2005, Okada et al. introduced their techniques for anatomical segmentectomy using “hybrid VATS,” which is defined as a minimally invasive approach under direct visualization with muscle-sparing minithoracotomy and video assistance [31]. The technical and oncological feasibility of this MIS approach for complex segmentectomy has been reported [26,32]. Several reports have demonstrated that complete VATS, which refers to a procedure that is performed primarily with a monitored view without direct visualization or rib spreading, and a recently emerged uniport VATS are technically feasible for complex segmentectomy [29,33,34]. However, their learning curves and generalizability remain unclear. Recently, the RATS approach has gained popularity in the field of general thoracic surgery, with several reported advantages over VATS, including ergonomic design, 3D-binocular vision, elimination of tremors, increased degree of motion, and enhanced manipulation. Such advantages are considered to enhance precise movement, surgeon view, and dexterity; and make MIS segmentectomy easier to adopt and perform [35,36,37].

### 2.2. Complex Segmentectomy for Single or Combined Basilar Segments

In general, individual segmentectomies of the basilar segment (a single or combined basilar segment less than the basilar segment) are more technically challenging than other individual segmentectomies [38]. The anatomical pattern of basilar segments is more complex than that of other lung structures [13]. Depending on the location of the tumor and the anatomical pattern of individual patients, resection of a single segment or combined adjacent segments can be planned. One of the aspects to determine the difficulty and complexity of segmentectomy is the number and anatomical configuration of the intersegmental planes that should be divided during the surgery. Among single basilar segmentectomies, S9 segmentectomy is more complex because more than three intersegmental planes should be divided during the procedure (S6–S9, S7–S9, S8–S9, and S10–S9 in the right; and S6–S9, S8–S9, and S10–S9 in the left). Moreover, the broncho–vascular branching pattern of S9 varies significantly. Therefore, tumors existing in the S9 segment are more frequently removed by combined segmentectomies, such as S8–S9 or S9–S10 segmentectomies, rather than by single S9 segmentectomy.

## 3. Planning and Navigation for Segmentectomy

Although the sequence of procedures for lobectomy and segmentectomy are similar, segmentectomy requires additional maneuvers, including the following: (1) identification and division [or preservation] of the segmental and/or subsegmental branches of the vessels and bronchi; (2) identification of and dissection along the intersegmental veins in order to appropriately divide the proximal intersegmental plane; and (3) identification and division of the peripheral intersegmental plane in order to obtain adequate surgical margins.

In decision-making during lung segmentectomy, thoracic surgeons should be confident of the following: (i) which branches should be divided or preserved; (ii) the appropriate sequential order for dividing segmental branches of the PA, PV, and bronchi; (iii) the margin distance can be obtained by a planned procedure; and (iv) the location of the intersegmental plane and tumor. These may be challenging for thoracic surgeons because of the wide variety of segmental/subsegmental branching patterns of the pulmonary vessels and bronchi with frequent anomalies. Therefore, preoperative planning and intraoperative navigation are useful in segmentectomy.

### 3.1. Surgery-Oriented Classification of Segmental Anatomy, and Preoperative Planning for Segmentectomy

To prepare for an anatomical segmentectomy, it is important to obtain detailed anatomical information and understand the spatial configuration of the anatomy of each patient preoperatively. Although there are infinite variations in hilar anatomy based on the number, origin, and anomalies of the segmental vascular and bronchial branches and their relationship, the hilar anatomy of each patient can be classified into several patterns [12,13,39,40]. 3DCT imaging based on volume-rendering reconstruction is useful for examining each segmental vascular and bronchial branch, determining the presence of anomalies, and classifying anatomical patterns preoperatively [17,18,19]. Preoperative planning and simulation of segmentectomy using 3DCT images is also helpful for surgeons to obtain preoperative consensus regarding the type and extent of resection. Importantly, specific surgical strategies are affected by individual anatomical patterns. Our group previously developed simplified 3D anatomical models of the right upper lobe [12], in which an approach to the intersegmental vein was determined according to the anatomical model, technical usefulness, and acceptable outcomes reported recently [41].

### 3.2. Intraoperative Navigation

Even after successful preoperative planning and simulation, challenging situations regarding the recognition of anatomy remain common during segmentectomy. Such issues include (1) discrepancies between the preoperative plan and the actual procedure due to misrecognition of 3DCT anatomy and/or actual anatomy; and (2) discrepancies between surgeons (e.g., primary operator and assistant). Intraoperative navigation with 3DCT images allows surgeons to discuss and/or confirm the segmental anatomy based on both 3DCT images and actual operative findings and perform safe and secure lung segmentectomy with confidence [14,25]. The use of additional monitors next to the surgeon, or a tablet device contained in the sterile bag has been reported for 3DCT visualization in intraoperative navigation [14,15].

### 3.3. Surgeon-Oriented Planning/Navigation Using a Novel 3DCT Software Dedicated for Lung Segmentectomy

3DCT-based anatomical evaluation in individual patients is useful for preoperative planning and intraoperative navigation for segmentectomy. However, there remain some potential drawbacks, as follows: (a) this is a time-consuming process that requires technical skill in preparing detailed 3D images for surgeons; (b) difficulty in developing 3DCT from CT scans without contrast; and (c) difficulty in obtaining more realistic simulation images, such as an image showing stumps of segmental branches with divided intersegmental planes and preserved intersegmental veins. Recently, volume-rendering reconstruction software dedicated to lung segmentectomy was approved in Japan (REVORAS, Ziosoft, Inc., Tokyo, Japan). This software has a segmentectomy planning function with the following surgeon-oriented features: (1) automated reconstruction of 3D images by either contrast or non-contrast CT scans (surgeons simply need to select a series of CT scans in a “one-click 3D reconstruction”); (2) semi-automated segmentectomy planning based on the location of the tumor and/or the dividing level of the bronchial or arterial branches (surgeons simply need to determine the dividing level); (3) semi-automated planning modification, such as changes in the level of stumps, selecting additional arterial or bronchial branches to be resected, and changes between dividing and preserving settings (surgeons simply need to determine them); (4) automated measurements of margin distance and resected volume/percentage; and (5) segmentectomy-simulated, surgeon-oriented specific 3D views that are useful for understanding the hilar structures that are being divided, are to be divided next, or are to be preserved, and understanding the shape of the lung parenchyma to be resected with the intersegmental plane and corresponding veins (Figure 1). The advantages and novelty of this segmentectomy planning by REVORAS are as follows: (i) an easy-to-use tool for surgeons; (ii) accepted use of non-contrast CT for “one-click 3D reconstruction”; (iii) separate segmentectomy planning based on the dividing bronchus or pulmonary artery, providing more reliable planning with an accurate resection margin according to the intraoperative identification technique of the intersegmental plane (ventilation-based identification fits for planning based on the bronchial stump, whereas perfusion-based identification fits for planning based on the arterial stump); and (iv) surgeon-oriented, intraoperatively useful 3D images with “key parts for segmentectomy”, including the bronchial and vascular stumps and intersegmental plane.

## 4. Localization of Small Tumors

Very small nodules or subsolid nodules detected by CT scan are often difficult to palpate, particularly when they exist deep in the lung parenchyma. One of the drawbacks of the MIS approach is the difficulty in palpating small nodules. In the RATS approach, lung palpation is essentially impossible. In the VATS approach, “one-finger” palpation may be possible, but bimanual palpation is impossible. Therefore, intraoperative tumor localization without palpation is of great importance, especially for MIS, and various localization strategies have been proposed. We have divided these strategies into three categories based on their use of markers and the approach for placement: (1) CT-guided percutaneous marker placement, (2) bronchoscopic marker placement, and (3) intraoperative ultrasonography without marker placement. The characteristics of each procedure are listed in Table 1. Important qualities in tumor localization are safety, technical feasibility, accurate intraoperative localization, and real-time monitoring of the tumor location.

### 4.1. Computed Tomography-Guided Percutaneous Marker Placement

To [21,45,46,47,48] localize small lung tumors, an intraoperatively identified marker can be placed under the guidance of a CT scan. Markers that have been used for this technique include a hookwire [42,43,44], microcoils [21,45,46,47,48], dyes [49,50,51,52], contrast media [53,54,55], and radioisotopes [56,57]. Technical limitations of CT-guided percutaneous marking include marking for tumors in specific locations, such as the apex; the posterior subpleural area just medial to the scapula; the deep area, including the hilum and the center of the lobes; the area in close proximity to the great vessels; and the subpleural area adjacent to the diaphragm. Given that the marker is inserted through the pleura and peripheral lung parenchyma, CT-guided percutaneous marker placement carries a risk of pneumothorax, intrapulmonary hemorrhage, hemothorax, and air embolism [44,46].

Hookwire placement is one of the most popular localization strategies [43]. A hookwire, which is a short wire with an anchor at its tip and a string attached to the end, is deployed thorough a needle under the guidance of a CT scan. The location of the marker and the relationship between the marker and the targeted tumor can be confirmed by CT scan after marker placement. Intraoperatively, the extrapleural extension of the wire or string attached to the wire can be easily identified through thoracoscopy; in most cases, no additional modalities for localization, such as fluoroscopy, are required, which is one of the advantages of this procedure. Although the most common complication associated with a hookwire procedure is dislodgement or migration of the hookwire after placement [42], this procedure is also associated with the risk of other complications, as described above. In particular, air embolism has been reported to occur almost exclusively following hookwire placement [73,74,75]. Horan et al. observed that the hookwire passed through a subsegmental bronchus in the resected specimen in a patient who had massive air embolism following a hookwire placement, and suggested that air could pass from the bronchiole to the adjacent vasculature through a placed hookwire, the risk of which may be higher given the longer trajectories necessary for deeper or more medially placed lesions [73].

A microcoil is another marker for tumor localization, which can be placed and preoperatively confirmed using a CT scan similar to the hookwire procedure [21,45,46,47,48]. In contrast to the hookwire procedure, intraoperatively placed microcoils are generally difficult to be seen by thoracoscopic view, and should therefore be identified with the use of fluoroscopy. Although CT-guided microcoil marking may carry a similar risk of complications to the hookwire procedure, some studies have suggested a lower risk of complications after CT-guided microcoil marking than after hookwire marking [46].

Several types of liquid materials, including dyes (e.g., methylene blue [49,50], indocyanine green (ICG) [51,52]), contrast media (e.g., lipiodol [53,54]), and radioisotopes (e.g., technetium 99 [56,57]) are used for localization under CT guidance. Methylene blue is visible under thoracoscopic inspection without any specific imaging system [76]. For intraoperative visualization and/or identification, and in contrast to methylene blue, ICG requires infrared fluorescence imaging systems, contrast media, such as lipiodol, require fluoroscopy, and radioisotopes require a gamma ray detector. In general, CT-guided placement of liquid materials carries a risk of spillage of agents into the pleural space, which may result in ineffective localization.

### 4.2. Bronchoscopic Marker Placement

Various types of markers for localization can be placed via bronchoscopy, including microcoils [58,59], dyes [60,61,62], and contrast media [63,64]. Guidance with fluoroscopy, CT, and/or virtual airway images can be used to select a target segmental bronchus and place a marker at an appropriate location [58,59,60,61,62,63,64]. Intraoperative identification of these markers is similar to that of markers placed with CT guidance. Compared to the CT-guided approach, the bronchoscopic approach generally has better access to tumors in relatively deep areas, apical areas, and basilar areas close to the diaphragm, where a needle cannot reach safely.

#### 4.2.1. Virtual-Assisted Lung Mapping (VAL-MAP)

VAL-MAP is an integrated technique for tumor localization to secure adequate margins, and consists of preoperative marker placement planning using 3DCT, placement of markers (multiple subpleural dye marks surrounding the tumor with or without intrabronchial microcoil in a deeper area than the tumor), 3DCT following marker placement, and surgical resection with visual identification of dyes and fluoroscopic identification of microcoils [62,77,78]. The original VAL-MAP technique did not include microcoil placement, but this was later added in VAL-MAP 2.0 to address the possibilities of inadequate deep margins based on early experiences [20].

#### 4.2.2. Radiofrequency Identification (RFID) Marking

RFID is a newly developed technique for tumor localization [22,65,66,67] has been approved for clinical use in Japan (SuReFInD, Hogy Medical Co., Ltd., Tokyo, Japan). This system consists of the following components: a) a micro RFID tag (3.2 × 1.6 × 0.8 mm) with a nickel-titanium coil anchor (IC tag) as a marker; b) a bronchoscopic delivery device that can pass through a 2 mm working channel of the bronchoscope; and c) a detection probe (10 mm in diameter) with a signal processing device. In clinical practice, the RFID localization technique involves the following four steps: (1) bronchoscopic placement of the marker into the peripheral bronchus within or adjacent to the tumor using a virtual bronchoscopic navigation technique under fluoroscopic or CT guidance within two days before surgery; (2) CT scan to confirm the exact location of the marker and the 3D relationship between the marker and the tumor; (3) intraoperative identification of the marker using a handheld, sterile detection probe; and (4) real-time monitoring of the tumor location when dividing the lung (Figure 2). The advantages of this procedure include: (i) the marker can stay in place for at least 48 h, (ii) multiple markers can be detected separately with a unique identification in each marker, and (iii) the location of the marker can be easily confirmed even when dividing the lungs (real-time monitoring). Possible drawbacks include the following: (1) although rare, dislodgement of the marker may occur when it is placed into a relatively large bronchus >3 mm in diameter; and (2) precise placement of markers depends on bronchoscopic skills and/or equipment, such as virtual or electromagnetic navigation [79,80].

### 4.3. Intraoperative Ultrasonography without Marker Placement

Peripheral lung nodules can be localized intraoperatively using a flexible ultrasonography probe following lung deflation [68,69,70,71,72]. Intraoperative ultrasonography has been reported to reliably detect peripheral small nodules as small as 2 mm in diameter and up to 2.4 cm in depth from the visceral pleura [69]. Several reports have suggested that ultrasonography would be useful not only for localization, but also for the differential diagnosis of tumors based on sonographic patterns. Ultrasonographic localization offers several advantages over other localization strategies, in that it does not require marker placement without an associated risk, such as pneumothorax, and provides real-time imaging without ionizing radiation when dividing the lungs. Possible drawbacks of ultrasonographic localization include its limited utility for deep lesions, limited mobility of the probe, operator-dependent image quality, and the fact that the mages are highly influenced by incomplete deflation and/or background diffuse lung diseases, such as emphysema.

## 5. Identification of the Intersegmental Plane

In addition to accurately identifying the tumor location, accurate identification of the intersegmental plane is crucial to achieve a successful segmentectomy with an adequate margin and decreased risk of air leak. Basically, the intersegmental plane is delineated intraoperatively using differences in ventilation or perfusion between the segment(s) to be resected and those to be preserved.

The conventional maneuver to identify the intersegmental plane is delineation of the inflation-deflation line by inflating the lungs following occlusion or division of the target segmental bronchus to obtain collapsed segment(s) to be resected and expanded segment(s) to be preserved [81]. However, in this maneuver, it is sometimes difficult to delineate a clear inflation-deflation line due to inflation of the deflated segment(s) to be resected because of collateral ventilation through the pores of Kohn; this has been suggested to be due to a relatively strong positive airway pressure into the segments to be preserved [32,82]. Tsubota reported a “reverse” maneuver, in which the whole lobe was inflated first before the inflation was stopped and the target segmental bronchus was ligated; this results in a continuous inflated target segment to be resected and deflated segments to be preserved [82]. Since the lungs are fully inflated in the above-mentioned maneuvers, expansion of the lungs limits the working space in the pleural cavity during thoracoscopic procedures, which makes delineation of the inflation-deflation line unreliable [23].

### 5.1. Selective Segmental Inflation

The selective segmental inflation technique, in which the segment to be resected is selectively inflated using bronchoscopic jet-ventilation, was first reported by Okada et al. in 2007 [32]. Modified or similar selective segmental inflation techniques have also been reported [83,84]. The working space is not compromised during MIS in these procedures owing to the limited expansion of the lungs. In addition, given that the intersegmental plane is clearly visible in the selective segmental inflation technique, surgeons may be able to divide the plane using electrocautery instead of a stapler, which may prevent compromised lung parenchyma without full expansion created by stapler application [32]. However, potential drawbacks of this procedure include the requirement of additional materials, such as jet ventilation, and the ability to selectively administer ventilation into the target segmental bronchus. Recently, a modified selective segmental inflation technique without jet ventilation has been reported, in which the segmental bronchus of the target segment was divided first, and then the whole lungs were re-expanded with positive continuous pressure, which expanded the target segment through the pores of Kohn, subsequent discontinuation of the positive pressure and waiting visualized the intersegmental plane with continuous expansion of the target segment [85].

### 5.2. Indocyanine Green

ICG can be used for pulmonary arterial perfusion-based identification of the intersegmental plane as an alternative to the above-described ventilation-based techniques during MIS. Misaki et al. first reported the systemic administration of ICG to identify the intersegmental plane using infrared thoracoscopy [86,87]. This technique has been applied in thoracic MIS [88,89]. The advantages of this procedure include its ease of use, lack of inflation of the lungs and preserved working space, and the availability of Firefly mode in the RATS procedure, which is an integrated fluorescence imaging system in the da Vinci Surgical System (Intuitive Surgical, Inc. Sunnyvale, CA, USA) that can be controlled by a console surgeon [37]. However, potential drawbacks of this procedure include the requirement of infrared fluorescence imaging systems, and potential poor visibility in patients with compromised parenchymal perfusion, such as in those with emphysema. In contrast to perfusion-based techniques by systemic dye injection, selective transbronchial injection of dyes, such as ICG [90] and methylene blue [91], have also been investigated for intraoperative identification of the intersegmental plane.

In terms of the accuracy of the identification of the intersegmental plane, no study has compared perfusion- and ventilation-based techniques, and this is therefore warranted in the future.

## 6. Precision Lung Segmentectomy in Shinshu University Hospital

To perform safe, secure, and precise lung segmentectomy, even for deep, small, and impalpable tumors, we adopted two novel technologies in Shinshu University Hospital: (1) surgeon-oriented 3DCT-based resection planning using a novel 3DCT processing software called “REVORAS,” and (2) tumor localization using a novel RFID technology called “SuReFInD.” The details of each procedure were described earlier. In this section, we demonstrate the utilization of these novel techniques in clinical practice (Figure 3).

In Shinshu University, the criteria for lung segmentectomy for malignant diseases are as follows: (1) ≤2 cm subsolid nodules with ≥50% ground-glass appearance on CT (≤50% consolidation/tumor ratio); (2) ≤2 cm nodules with a long doubling time (>400 days) in radiological surveillance; (3) multifocal lung cancer; (4) elderly or high-risk candidates; and (5) metastatic lung tumors that are not suitable for wedge resection because of the location or size of the tumors. For all the patients who undergo segmentectomy, we create 3DCT images for the attending surgeons and/or trainees to perform segmentectomy planning. Based on the planning with expected volume loss and margin distance, we determine which segment(s) and/or subsegment(s) should be removed, and which segmental/subsegmental branches should be divided or preserved. Then, we determine whether tumor localization is required for the planned segmentectomy with adequate margins. Among the localization procedures described earlier, we prefer the RFID technique because of its safety and availability for real-time monitoring when dividing the lung parenchyma. In cases with a peripheral tumor without an accessible target bronchus adjacent to the tumor, we utilize the hookwire technique. During surgery, we visualize 3DCT images for segmentectomy planning for intraoperative navigation using two additional monitors next to the surgeons. We also employ the TilePro function in RATS segmentectomy to visualize 3D images.

To identify the intersegmental plane, we employ either the selective segmental ventilation technique using jet ventilation or the systemic administration of ICG, depending on the situation. However, in cases undergoing complex segmentectomies, especially in single (or combined) segmentectomy of the basilar segment, or RATS segmentectomies, we prefer ICG technique rather than selective segmental ventilation because, in general, the target segmental bronchus should be divided after the division of segmental vessels during selective segmental ventilation technique to divide the intersegmental plane immediately after jet ventilation, which would be difficult in such cases.

C/T, consolidation-tumor; RFID, radiofrequency identification; VDT, volume doubling time; 3DCT, three-dimensional computed tomography.

## 7. Limitations

This review focused on recent technical advances in lung segmentectomy, especially via minimally invasive technology. We attempted to demonstrate up-to-date technologies with published evidence as much as possible. However, several characteristics of the techniques described in this review are not supported by scientific evidence. Furthermore, we did not compare segmentectomy and the current standard treatment, lobectomy, in terms of postoperative oncologic and physiologic outcomes, as this was beyond the scope of this review article. However, a recent large randomized trial (JCOG0802) will soon provide evidence-based comparative data between segmentectomy and lobectomy.

## 8. Conclusions

Here, we reviewed published evidence in MIS segmentectomy, with a particular focus on preoperative planning, intraoperative navigation, and tumor localization in patients with early-stage small lung cancer. In this era of personalized cancer treatment, thoracic surgeons should be familiar with technical advances in MIS segmentectomy to provide safe and secure “precision segmentectomy” based on individual tumor location, lung function, and anatomy.

## Figures and Tables

**Figure 1 cancers-13-03137-f001:**
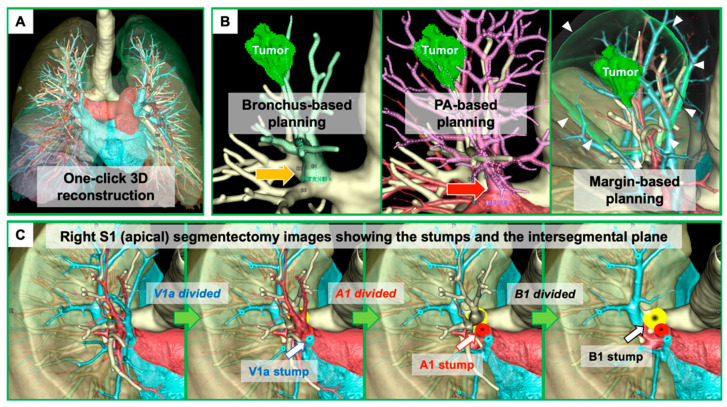
Three-dimensional computed tomography (3DCT) images in segmentectomy planning using a novel 3DCT processing software (REVORAS, Ziosoft, Inc., Tokyo, Japan). (**A**) Three-dimensional (3D) reconstruction can be obtained by the surgeon with “one click” to select a series of CT scans with or without contrast. (**B**) There are various types of segmentectomy planning, including bronchus-based, pulmonary artery (PA)-based, and margin-based. Yellow arrow: level of division of the target bronchus. Red arrow: level of division of the target PA. White arrowheads; two-centimeter margin from the tumor. (**C**) Simulation images, including the bronchial and vascular stumps and the intersegmental plane.

**Figure 2 cancers-13-03137-f002:**
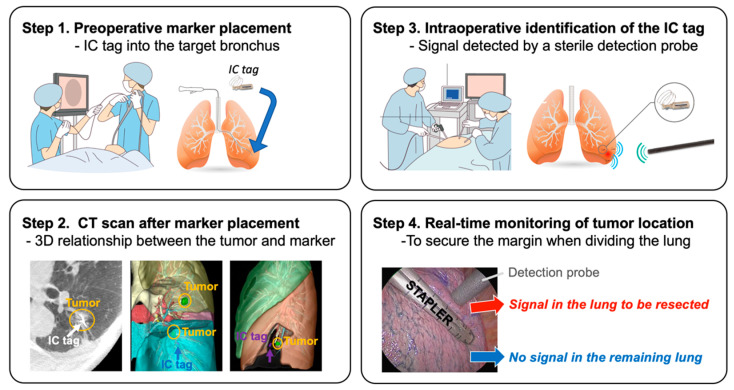
Schema of four steps in radiofrequency identification (RFID) marking. Step (1): Bronchoscopic placement of the RFID marker, IC tag, into the peripheral bronchus within or adjacent to the tumor using a virtual bronchoscopic navigation technique. Step (2): Computed tomography scan to confirm the exact location of the marker and the three-dimensional relationship between the marker and tumor. Step (3): Intraoperative identification of the marker using a handheld sterile detection probe. Step (4): Real-time monitoring of the tumor location when the lung parenchyma is divided.

**Figure 3 cancers-13-03137-f003:**
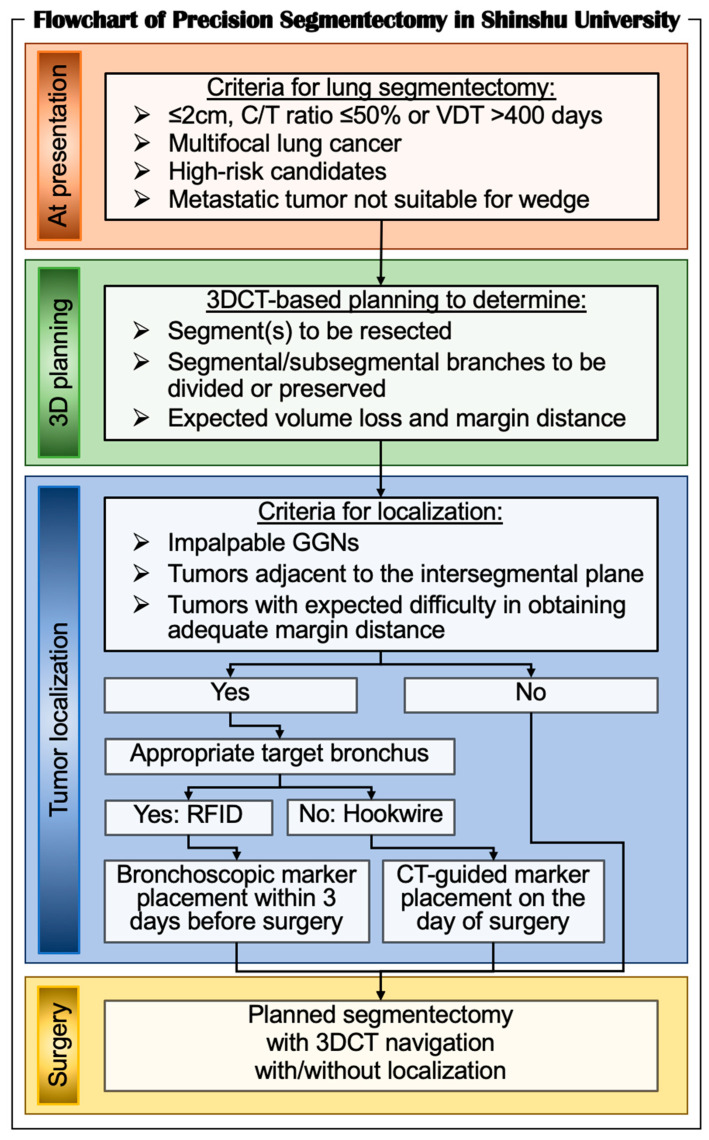
Flowchart of precision segmentectomy in Shinshu University.

**Table 1 cancers-13-03137-t001:** Characteristics of the localization procedures.

	Markers	Preoperative Confirmation	Intraoperative Detection	Access to Deep Lesion	Multiple markers ^#^	Real-Time Monitoring *	Pneumo-Thorax	Air Embolism	Other Complications
CT-guided percutaneous approach									
Hookwire [42,43,44]	Hookwire	CT	Visual (string)	Difficult ^¶^	No	No	Yes (38%) [44]	Yes (0.6%) [44]	Dislocation, hemorrhage
Microcoil [21,45,46,47,48]	Microcoil	CT	Fluoroscopy, CBCT	Difficult ^¶^	No	No	Yes (70%) [45]	Unknown ^§^	Dislocation
Dye [49,50,51,52]	Methylene blue, indigo carmine, ICG	N/A	Visual (dye)	Difficult ^¶^	No	No	Yes (20%) [51]	Unknown ^§^	Limited retention with diffusion, pleural spillage
Contrast media [53,54,55]	Lipiodol, barium	CT	Fluoroscopy, CBCT	Difficult ^¶^	No	No	Yes (17%) [55]	Unknown ^§^	Hemosputum
Radioisotope [56,57]	Technetium 99	Nuclear scintigram	Gamma probe	Difficult ^¶^	No	Yes	Yes (8%) [57]	Unknown ^§^	Pleural spillage
Bronchoscopic approach									
Microcoil [58,59]	Microcoil	CT	Fluoroscopy, CBCT	Relatively easy ^¶^	No	No	Unknown ^§^	Unknown ^§^	Dislocation
Dye [60,61,62]	Methylene blue, indigo carmine, ICG	N/A	Visual (dye)	Relatively easy ^¶^	No	No	Unknown ^§^	Unknown ^§^	Limited retention with diffusion
VAL-MAP [20]	Indigo carmine (and microcoil)	CT	Visual (dye), fluoroscopy, CBCT	Relatively easy ^¶^	No	No	Unknown ^§^	Unknown ^§^	Hemorrhage
Contrast media [63,64]	Barium	CT	Fluoroscopy, CBCT	Relatively easy ^¶^	No	No	Unknown ^§^	Unknown ^§^	Inflammatory change to barium
RFID [22,65,66,67]	RFID tag	CT	RFID probe, fluoroscopy, CBCT	Relatively easy ^¶^	Yes	Yes	Unknown ^§^	Unknown ^§^	Dislocation
Imaging guidance without marker									
Ultrasonography [68,69,70,71,72]	No marker	N/A	US	Difficult ^¶^	N/A	Yes	No	No	

CBCT: Cone-beam computed tomography, CT: Computed tomography, ICG: Indocyanine green, N/A: Not applicable, RFID: Radiofrequency identification, US: Ultrasonography, VAL-MAP: Virtual-assisted mapping. ^#^ Each marker can be differentiated from other markers with unique identification. * Real-time confirmation that the marker exists in the lung to be resected without using fluoroscopy when dividing the lung parenchyma. ^¶^ Based on the authors’ subjective opinions without available scientific data. ^§^ No published incidence available.

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
