# Peer review of "Technical Advances in Segmentectomy for Lung Cancer: A Minimally Invasive Strategy for Deep, Small, and Impalpable Tumors"

_cancers, 2021, doi:10.3390/cancers13133137_

Round 1
Reviewer 1 Report
I think that the review is very good. However, I would add a paragraph for planning segmentectomy for basal segments (which are easy to combine, which basal segments can be done in singular manner).
Also, please also discuss the potential pitfalls of overly aggressive segmentectomy (e.g. diminishing returns on PFTs, air leaks, torsion, and oncologic safety). Please also discuss where preoperative IR guided Bx could be potentially a better alternative.
We use CT guided mucus seal "plug" that is placed on the lung surface, that is associated with low complication rate..
Author Response
We thank reviewer #1 for their suggestions and comments. Please find our specific responses
below.
1. I think that the review is very good. However, I would add a paragraph for planning
segmentectomy for basal segments (which are easy to combine, which basal segments can be
done in singular manner).
Response: We thank the reviewer for this suggestion. We created an additional paragraph on
“complex segmentectomy for single or combined basilar segments” to describe the topic.
2.2 Complex segmentectomy for single or combined basilar segments (page 3, lines 105-119
[“No Markup” function])
In general, individual segmentectomies of the basilar segment (a single or combined basilar
segment less than the basilar segment) are more technically challenging than other individual
segmentectomies. The anatomical pattern of basilar segments is more complex than that of
other lung structures. Depending on the location of the tumor and the anatomical pattern of
individual patients, resection of a single segment or combined adjacent segments can be
planned. One of the aspects to determine the difficulty and complexity of segmentectomy is
the number and anatomical configuration of the intersegmental planes that should be divided
during the surgery. Among single basilar segmentectomies, S9 segmentectomy is more
complex because more than three intersegmental planes should be divided during the
procedure (S6-S9, S7-S9, S8-S9, and S10-S9 in the right; and S6-S9, S8-S9, and S10-S9 in the
left). Moreover, the broncho-vascular branching pattern of S9 varies significantly. Therefore,
tumors existing in the S9 segment are more frequently removed by combined
segmentectomies, such as S8-S9 or S9-S10 segmentectomies, rather than by single S9
segmentectomy.
2. Also, please also discuss the potential pitfalls of overly aggressive segmentectomy (e.g.
diminishing returns on PFTs, air leaks, torsion, and oncologic safety). Please also discuss
where preoperative IR guided Bx could be potentially a better alternative.
Response: We thank the reviewer for this suggestion. Despite oncologic concerns and a
potentially higher complication risk, including prolonged air leakages, in segmentectomy
compared with lobectomy, which is the current standard surgery, the use of segmentectomy
has been increasing. Data from a large, multi-institutional, prospective randomized trial
assessing the outcomes of segmentectomy and lobectomy will soon enable surgeons to select
an appropriate surgical approach with evidence-based advantages and drawbacks of
segmentectomy. However, a discussion about this point is beyond the scope of this paper; we
have described this in the limitations section.
7. Limitations (pages 5-6, lines 417-425 [“No Markup” function])
This review focused on recent technical advances in lung segmentectomy, especially via
minimally invasive technology. We attempted to demonstrate up-to-date technologies with
published evidence as much as possible. However, several characteristics of the techniques
described in this review are not supported by scientific evidence. Furthermore, we did not
compare segmentectomy and the current standard treatment, lobectomy, in terms of
postoperative oncologic and physiologic outcomes, as this was beyond the scope of this
review article. However, a recent large randomized trial (JCOG0802) will soon provide
evidence-based comparative data between segmentectomy and lobectomy.
3. We use CT guided mucus seal "plug" that is placed on the lung surface, that is associated with
low complication rate.
Response: We thank the reviewer for sharing the information, and we apologize for the lack of
information on the CT-guided mucus seal plug procedure. We should have incorporated the
procedure in this revised manuscript; unfortunately, we could not find literature in the PubMed
database using the words “mucus,” “seal,” or “plug.”
Reviewer 2 Report
I have read with great interest a paper entitled “Technical advances in segmentectomy for lung cancer: A minimally invasive strategy for deep, small, and impalpable tumors”. Authors concentrate on an important topic of technical advances in minimally invasive segmentectomies in early lung cancer. Recent advances in imaging techniques enable sophisticated assessment of the anatomy of vascular structures making the resection better planned. Moreover, in the case of impalpable tumor techniques of marking the tumor are presented. An interesting aspect of the study is the description of the technique of marking the borders of the segment. This review paper brings a number of important issues supported by a significant number of recently published papers. The quality of used English language used is good.
Despite the fact that the authors made significant efforts preparing the paper I think the paper is too superficial for publication in Cancers. The mentioned techniques are not systematically listed. There is no exact comparison of papers’ limitations and advantages. The listed methods in Table 1 are not supported by references. I believe that this kind of comparison deserves rather a systematic review or a meta-analysis. This kind of narrative review may be more interesting in the journal with narrower scope in the field of general thoracic surgery or thoracic oncology. Maybe splitting the paper into parts and presenting a more detailed approach would gain scientific soundness.
In conclusion, I regret to recommend rejection to this form of presentation of this interesting issue.
Author Response
We thank reviewer #2 for their suggestions and comments. Please see our response in attach.

Reviewer 3 Report
Eguchi and associates present a Review article outlining important recent technical advances that enhance the performance of anatomic segmentectomy. Anatomic segmentectomy is emerging as a valid alternative to lobectomy in the management of small tumors. This approach is now routinely being performed utilizing minimally-invasive approaches. Several challenges related to variable segmental anatomy, detection of the intersegmental plane and localization of small, otherwise non-palpable, tumors are appropriately discussed in this timely Review.
The article is concise, and well-written. The authors provide a fair and succinct summary of the importance of 3D anatomic rendering to assist in preoperative and intraoperative planning. Additional detail on how to perform/use the 3D localization techniques mentioned might be instructive.
In the introduction, it is worth mentioning the recently presented results of the JCOG0802 study suggesting that outcomes following segmentectomy are comparable to lobectomy, and that there may even be an advantage in survival with segmentectomy.
What is the authors' preferred localization method, and why?
Author Response
We thank reviewer #3 for their suggestions and comments. Please find our specific responses in attach.

Reviewer 4 Report
This is a well-written review on pulmonary segmentectomy for early stage lung cancer. The author emphasize on technologies available for preoperative planning, intraoperative identification of small and deep tumors and intraoperative techniques to define intersegmental planes. All are very valuable for surgeons performing pulmonary segmentectomies, especially those of the "complex' types
Author Response
We thank reviewer #4 for their suggestions and comments. Please find our specific responses
below.
1. This is a well-written review on pulmonary segmentectomy for early stage lung cancer. The
author emphasize on technologies available for preoperative planning, intraoperative
identification of small and deep tumors and intraoperative techniques to define intersegmental
planes. All are very valuable for surgeons performing pulmonary segmentectomies, especially
those of the "complex' types
Response: We thank and appreciate their recognition and pertinent comments.
Round 2
Reviewer 2 Report
I had the pleasure of re-reviewing the paper entitled: Technical advances in segmentectomy for lung cancer: A minimally invasive strategy for deep, small, and impalpable tumors submitted for publication in Cancers. My previous review was rigorous. I suggested rejecting the paper indicating a number of shortcomings. I would like to appreciate the efforts made to improve the quality of the submitted paper. The significant improvements in the text made it more informative. The redesigned Table 1 is only a part of essential betterment. The added paragraphs about the local approach in Shinshu University, study limitations, and complex metastasectomies inclined me to change the previous decision. I recommend accepting the paper.